# Knowledge Representation Combining Quaternion Path Integration and Depth-wise Atrous Circular Convolution

**Xinyuan Chen**[1,2]        **Zhongmei Zhou**[1]        **Meichun Gao**[1]        **Daya Shi**[1]        **Mohd Nizam Husen**[2]

[1]School of Technology, Fuzhou Technology and Business University, Fuzhou, Fujian, China
[2]Malaysian Institute of Information Technology, Universiti Kuala Lumpur, Kuala Lumpur, Malaysia,

## Abstract

Knowledge models endeavor to improve representation and feature extraction capabilities while keeping low computational cost. Firstly, existing embedding models in hypercomplex spaces of non-Abelian group are optimized. Then a method for fast quaternion multiplication is proposed with proof, with which path semantics are computed and further integrated with the attention mechanism based on the idea semantic extraction of relation sequences could be regarded as a multiple rotational blending problem. A depth-wise atrous circular convolution framework is set up for better feature extraction. Experiments including Link Prediction and Path Query are conducted on benchmark datasets verifying our model holds advantages over state-of-the-art models like Rotate3D. Moreover, the model is tested on a biomedical dataset simulating real-world applications. An ablation study is also performed to explore the effectiveness of different components.

## 1 INTRODUCTION

Knowledge Graph (KG) is composed of structured fact triples. Entities in the triples are represented as nodes in the graph and the relations between head and tail entities are represented as edges connecting the nodes. KGs are widely applied in areas such as question answering [Hao et al., 2017] and personalized recommendation [Guo et al., 2020]. However, existing KGs are incomplete and contain noise. One of the ideas is to embed entities / relations into low-dimensional vector spaces and apply KG completion techniques to predict missing facts. For example, in RotatE [Sun et al., 2019] relations are mapped as rotations and the distances between the head vectors after rotations and the tail vectors are utilized to determine whether triple facts

are true. However, different KGs contain various proportions of multiple relation modes including Symmetry, Antisymmetry, Inversion and Composition. Different models are capable of learning representations for different modes and so far there is no perfect embedding solution.

Most of existing embedding models learn representations in the two smallest domains of Divisor Algebra, $R$ and $C$. With Quaternion algebra $H$ and Octonion algebra $O$ models could develop higher expressivity with less parameters; what's more, the characteristic of non-commutative law in non-Abelian groups helps to model the compositional relation mode. Three-dimensional (3D) and four-dimensional (4D) spatial embeddings with quaternions are adopted in Rotate3D [Gao et al., 2020] and QuatE [Zhang et al., 2019] respectively. Subsequent models enhance expressivity by adding entity / relation-specific quaternions or increasing embedding dimensions; larger parameter scales as well as limited feature extraction capabilities leave space for improvement.

Semantic information carried by relation paths between entity pairs helps to determine the validity of triples in knowledge inference. Many models employ frameworks including Recurrent Neural Network (RNN) [Jozefowicz et al., 2015], Long Short-Term Memory (LSTM) [Greff et al., 2016][Zhou et al., 2016] and Gated Recurrent Unit (GRU) [Lu and Duan, 2017] to merge vector sequences while the computational efficiency could be further boosted.

ConvE [Dettmers et al., 2018] performs 2-dimensional (2D) reshaping on concatenation matrices to enhance interactions and extracts deep non-linear features with Convolutional Neural Network (CNN). InteractE [Vashishth et al., 2020] adopts an optimized reshaping strategy as well as the circular convolution. In order to capture rich features of complex relations, inspired by neurons with different sizes of receptive fields, Atrous Convolution [Chen et al., 2017] expands the fields for larger interaction spaces while maintaining parameter scales. Moreover, introduction of the attention mechanism to integrate features extracted by

*Accepted for the 38th Conference on Uncertainty in Artificial Intelligence* (UAI 2022).

kernels with various sizes helps to stabilize model performance. The strategies above are jointly applied in our study.

APAC (A Knowledge Representation Model based on the Non- Abelian groups, Path Semantics and Depth-wise Atrous Circular Convolution) is proposed and main work includes:

1. A hypercomplex embedding model with improved score function and loss function designs is brought forward based on state-of-the-art (SOTA) quaternion models, in which Quaternion algebra, a Hamilton group with the smallest order is employed to learn multiple relational modes between entities. Embedding is also extended to the octonion space.

2. Based on the idea of multi-hop reasoning in Rotate3D, a fast multiplicative calculation method for quaternion sequences is proposed with proof for rapid feature mergers of relational paths which are then integrated with the attention mechanism.

3. A depth-wise atrous circular convolution framework is set up to enhance the feature extraction capability.

4. Experiments including Link Prediction and Path Query are carried out on benchmark and industry datasets to verify model effectiveness. Ablation study is further performed (see supplementary materials).

## 2   RELATED WORK

Embedding models could be roughly divided into translation/rotation-based distance models and similarity-based semantic models.

TransE [Bordes et al., 2013], a distance model, maps the relations to translation vectors. TransE holds that if a triple is valid, the head vector after translation should be close to the tail, denoted as

$$h + r \approx t, \tag{1}$$

where $h, r, t$ are the vector representations of the head entity, the relation and the tail entity respectively. L1 / L2 distance between vectors is taken as the score of the triple and a margin-based loss function is applied. With simple structure TransE achieves brilliant performance; however, it lacks the ability to learn symmetric relational representations. Most subsequent models improve by adding dimensions or expanding mapping spaces [Wang et al., 2014][Lin et al., 2015] followed by initiatives employing sparse matrix decomposition [Ji et al., 2016] to reduce the number of parameters. RotatE models relations as 2D rotations from head to tail entities in complex spaces with Hadamard product and normalized constraints. The calculation complies with the commutative law. Therefore, RotatE may not perform well on non-commutative relational modes (e.g., Adam's father's wife is not Adam's wife's father).

RESCAL [Nickel et al., 2015], an early semantic model, calculates the factorization of third-order adjacency tensors as triple scores. RESCAL holds strong expressivity but high complexity makes it difficult to train. DistMult [Yang et al., 2014] represents the relations as diagonal matrices to simplify calculations, capable of learning the symmetric and inverse modes. ComplEx [Trouillon et al., 2016] further extends the embedding to complex spaces to enhance the learning ability for anti-symmetric patterns. Hermitian product is used to calculate triple scores and reduce the number of parameters; however, it is still difficult for ComplEx to learn the non-commutative pattern. Lacroix et al. [2018] upgrade ComplEx with L3 regularization and a multi-class log loss.

Since hypercomplex spaces of non-Abelian groups hold strong expressivity and quaternion / octonion calculations are rather efficient, some models expand mapping spaces with them in recent years. Hyperbolic spaces are also taken into consideration [Chami et al., 2020].

QuatE represents relations as rotations in 4D spaces with quaternions to provide more degrees of freedom while avoiding Gimbal Lock. QuatE first calculates the Hamilton product between the head quaternion $Q_h$ and the unit relation quaternion $\hat{W}_r$, and then calculates the inner product of the result with the tail quaternion $Q_t$ so as to obtain the triple score. Compared with real/complex space models, QuatE could also learn symmetric (set the coefficients of imaginary parts to 0), anti-symmetric (conjugate quaternion), and inverse (coefficients set to -1) relation modes while enjoying larger spaces, less parameters and lower computational cost. On such basis, QuatDE [Gao et al., 2021], QuatRE [Nguyen et al., 2020] and DualE [Cao et al., 2021] further enhance the expressivity by increasing dimensions or adding quaternions, though limiting model scalability.

Most models above ignore rich semantic information contained in relation paths [Wang et al., 2016]. Lao et al. [2011] generate paths with the Random Walk algorithm and verify path values in knowledge inference. However, early studies take paths as atomic features, leading to huge feature matrices [Shang et al., 2019]. Neelakantan et al. [2015] and Das et al. [2016] decompose the paths into relation sequences and input them into RNN, reducing computational cost with parameter sharing. Nonetheless, the possibility that multiple paths to different extents associate with the candidate relations is ignored [Xie et al., 2017]. To solve this, Jiang et al. [2017] introduces the attention mechanism. Rotate3D models path-based multi-hop reasoning as multiple rotations and in our study a calculation method for fast rotation blending and integration is proposed.

Another vital indicator of knowledge representation models is feature extraction capability besides expressivity and computational overhead. Parameters in CNN are much less than those in fully connected neural networks and are

widely employed in Natural Language Processing in recent years. Compared with distance models, 2D convolution in ConvE is able to enhance interactions between entities/relations and extract richer features for embedding learning [Balažević et al., 2019a]. However local features are partly lost since ConvE leaves out translational/rotational attributes. Vashishth et al. [2020] believe that both the distance and semantic models could only capture shallow features, so they propose Checkered Reshaping and Circular Convolution to improve interactions. On this basis, Wang et al. [2021] suggest the multi-size atrous convolution combined with the attention mechanism could bring similar effects.

Therefore, in our model hypercomplex embedding is employed with optimization. Path semantics are extracted and integrated by fast rotational blending calculation and the attention mechanism. Also a depth-wise atrous circular convolution is defined to facilitate feature extraction.

## 3 APAC FRAMEWORK

As is shown in Figure 1. , APAC learns entity / relation embeddings in hypercomplex spaces. A fast relational path multiplication calculation is designed and the attention mechanism is introduced to integrate path semantics. The feature extraction capability is further enhanced with the depth-wise atrous circular convolution.

### 3.1 HYPERCOMPLEX EMBEDDING

Relation representations as rotations of 3D vectors in 3D subspaces of 4D spaces with quaternion embedding could effectively model multiple relational modes. QuatE's inner product score function is mostly used to solve logistic regression problems while distance-based score functions with L1 / L2 norm and margin-based normalized loss functions perform better with noise. Therefore, the Hamilton product between the head $Q_h$ and the relation r is firstly calculated, and then the distance between the result and the tail $Q_t$ is computed. The score function for quaternions is denoted as

$$\phi(\boldsymbol{h}, \boldsymbol{r}, \boldsymbol{t}) = \|\boldsymbol{h} \otimes \boldsymbol{r} - \boldsymbol{t}\|. \tag{2}$$

Accordingly, the loss function is defined as

$$L = -\log \sigma(\gamma - \phi(\boldsymbol{h}, \boldsymbol{r}, \boldsymbol{t})) - \\ \sum_{i=1}^{m} p\left(\boldsymbol{h}_i', \boldsymbol{r}, \boldsymbol{t}_i'\right) \log \sigma\left(\phi\left(\boldsymbol{h}_i', \boldsymbol{r}, \boldsymbol{t}_i'\right) - \gamma\right) + \lambda_1 \|\boldsymbol{Q}\|_2^2 + \\ \lambda_2 \|\boldsymbol{R}\|_2^2 \tag{3}$$

consulting the self-adversarial negative sampling in RotatE, where $\sigma$ is the Sigmoid function, $\gamma$ is the margin with a slack coefficient [Nayyeri et al., 2020], $\left(\boldsymbol{h}_i', \boldsymbol{r}, \boldsymbol{t}_i'\right)$ refers to the $i$th invalid triple, $m$ is the total number of invalid triples, $\lambda_1, \lambda_2$ are the coefficients for L2 norm en-

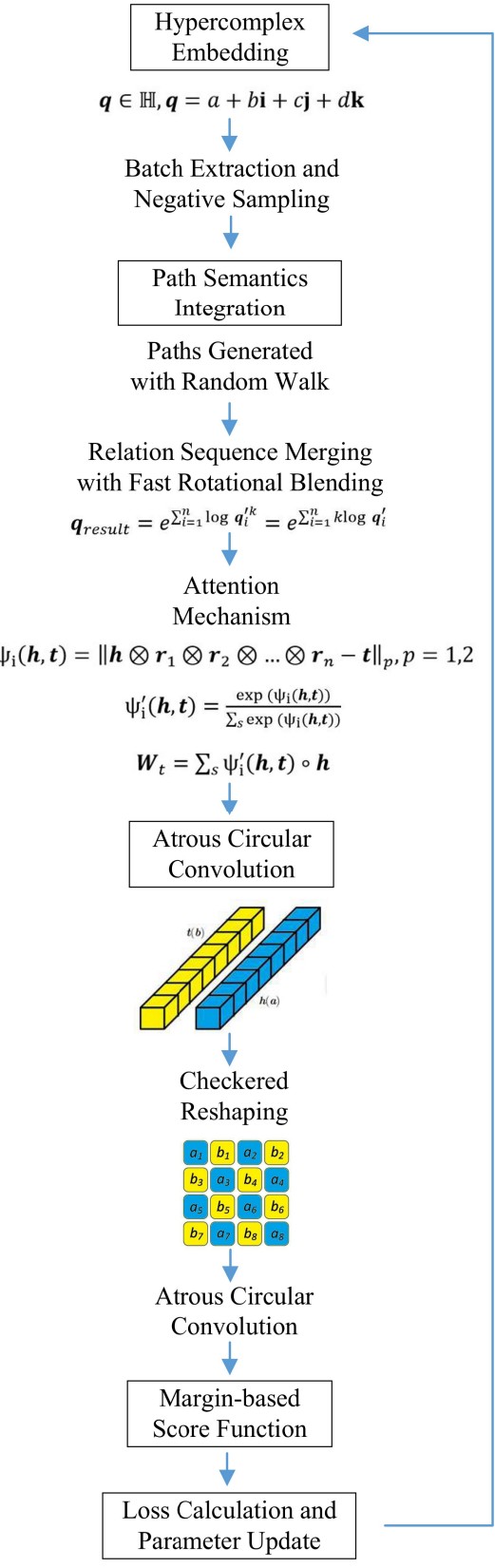

Hypercomplex Embedding

$$\boldsymbol{q} \in \mathbb{H}, \boldsymbol{q} = a + b\mathbf{i} + c\mathbf{j} + d\mathbf{k}$$

Batch Extraction and Negative Sampling

Path Semantics Integration

Paths Generated with Random Walk

Relation Sequence Merging with Fast Rotational Blending

$$\boldsymbol{q}_{result} = e^{\sum_{i=1}^{n} \log q_i'^k} = e^{\sum_{i=1}^{n} k \log q_i'}$$

Attention Mechanism

$$\psi_i(\boldsymbol{h}, \boldsymbol{t}) = \|\boldsymbol{h} \otimes \boldsymbol{r}_1 \otimes \boldsymbol{r}_2 \otimes \ldots \otimes \boldsymbol{r}_n - \boldsymbol{t}\|_p, p = 1,2$$

$$\psi_i'(\boldsymbol{h}, \boldsymbol{t}) = \frac{\exp(\psi_i(\boldsymbol{h}, \boldsymbol{t}))}{\sum_s \exp(\psi_i(\boldsymbol{h}, \boldsymbol{t}))}$$

$$\boldsymbol{W}_t = \sum_s \psi_i'(\boldsymbol{h}, \boldsymbol{t}) \circ \boldsymbol{h}$$

Atrous Circular Convolution

Checkered Reshaping

Atrous Circular Convolution

Margin-based Score Function

Loss Calculation and Parameter Update

Figure 1: Framework of APAC.

tity/relation constraints, $p\left(\boldsymbol{h}_i', \boldsymbol{r}, \boldsymbol{t}_i'\right) = \frac{\exp\beta\mathrm{f}\left(\boldsymbol{h}_i', \boldsymbol{t}_i'\right)}{\sum_{i=1}^{m}\exp\beta\mathrm{f}\left(\boldsymbol{h}_i', \boldsymbol{t}_i'\right)}$ calculates the probability distribution of negative sampling, $\beta$ is the sampling temperature, and $\mathrm{f}\left(\boldsymbol{h}_i', \boldsymbol{t}_i'\right) = -\phi\left(\boldsymbol{h}_i', \boldsymbol{t}_i'\right)$.

Similarly, the score function and loss function for octonions is defined similarly.

Since overall constraints are already imposed on $\boldsymbol{Q}$ and $\boldsymbol{R}$, the unit constraint on the relational quaternions seems unnecessary, compressing the entity rotation spaces and weakening expressivity. L1 normalization helps to generate sparse representations and only retain key features so as to reduce noise interference, but data could be left out on valuable channels. It is found that L2 constraint performs better in experiments.

## 3.2 EXTRACTION AND INTEGRATION OF PATH SEMANTICS

Following the path query solution proposed by Gao et al. [2020] based on multi-hop reasoning, in our study the multi-hop reasoning is regarded as continual Hamilton products of relational sequences taking advantages of the non-commutative characteristic of quaternions. For such calculation, a fast path feature extraction and the integration method is proposed.

Firstly, multiple paths are generated between entities with Random Walk and encoded as quaternion relation sequences. Length of path is defined as the number of relations in the path. Allow different length but set an upper threshold. Path feature extraction could be taken as multiple rotational blending. The operation space for rotations is non-linear and it is not right to directly add the rotational quaternions together.

**Theorem 1**: For a quaternion sequence, $\boldsymbol{q}_1, \boldsymbol{q}_2, \ldots \boldsymbol{q}_i, \ldots, \boldsymbol{q}_n, i = 1, 2, \ldots, n$, the continual Hamilton product could be calculated as

$$\boldsymbol{q}_{\text{result}} = e^{\sum_{i=1}^{n}\log q_i'^k} = e^{\sum_{i=1}^{n}k\log q_i'}. \quad (4)$$

Please see Appendix A for Proof and Illustration. Path score is denoted as

$$\Psi_{\mathrm{i}}(\boldsymbol{h}, \boldsymbol{t}) = \|\boldsymbol{h}\otimes\boldsymbol{r}_1\otimes\boldsymbol{r}_2\otimes\ldots\otimes\boldsymbol{r}_n - \boldsymbol{t}\|_p, p = 1, 2. \quad (5)$$

In order to reduce noise and extract key features, the attention mechanism is introduced to integrate path representations. denoted as

$$\Psi_{\mathrm{i}}'(\boldsymbol{h}, \boldsymbol{t}) = \frac{\exp\left(\phi_{\mathrm{i}}(\boldsymbol{h}, \boldsymbol{t})\right)}{\sum_s\exp\left(\phi_{\mathrm{i}}(\boldsymbol{h}, \boldsymbol{t})\right)}, \quad (6)$$

in which $s$ is the path set. The Softmax function is employed to normalize path scores. Structured representation

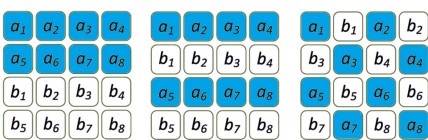

Figure 2: Stacked (left), Alternate (middle) and Checkered Reshaping (right).

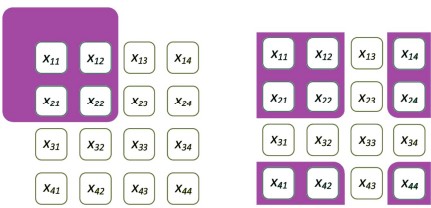

Figure 3: Ordinary Convolution (left) and Circular Convolution (right).

from path semantics of the tail entity is

$$\boldsymbol{W}_t = \sum_s \psi_{\mathrm{i}}'(\boldsymbol{h}, \boldsymbol{t}) \circ \boldsymbol{h}. \quad (7)$$

## 3.3 DEPTH-WISE ATROUS CIRCULAR CONVOLUTION

In this part the checkered reshaping and the depth-wise atrous circular convolution are combined to improve the model's capability in feature extraction.

The reshaping function is defined as $\pi : \boldsymbol{H}^k \times \boldsymbol{H}^k \rightarrow \boldsymbol{H}^{m\times n}$, where $m \times n = 2k$. The comparison of stacked, alternate and checkered reshaping is shown in Figure 2. . Vashishth et al. [2020] argue that entity/relation interactions could be divided into two types, heterogeneous and homogeneous, denoted as $\mathcal{N}_{\text{het}}(\pi, k)$ and $\mathcal{N}_{\text{homo}}(\pi, k)$, $\mathcal{N}_{\text{het}}(\pi, k) + \mathcal{N}_{\text{homo}}(\pi, k) = 2\begin{pmatrix}k^2\\2\end{pmatrix}$, $\mathcal{N}_{\text{het}}(\Omega_c(\pi), k)$ is with greater value in exploring entity/relation association. They prove that the proportion of $\mathcal{N}_{het}(\pi, k)$ is the highest with the checkered reshaping. Therefore, such strategy is applied.

Comparison between the ordinary convolution and the circular convolution is shown in Figure 3. . Vashishth et al. [2020] believe $\mathcal{N}_{het}(\Omega_c(\pi), k) \geq \mathcal{N}_{het}(\Omega_0(\pi), k)$ where $\Omega_c$ is the circular convolution and $\Omega_0$ is the ordinary convolution. The former is employed in our study, defined as

$$[\boldsymbol{I} \star \boldsymbol{\omega}]_{u,t} = \sum_{i=-\lfloor p/2\rfloor}^{\lfloor p/2\rfloor}\sum_{j=-\lfloor p/2\rfloor}^{\lfloor p/2\rfloor} \boldsymbol{I}_{[u-i]_m, [t-j]_n}\boldsymbol{\omega}_{i,j}, \quad (8)$$

where $\boldsymbol{I} \in \boldsymbol{H}^{m\times n}, \boldsymbol{\omega} \in \boldsymbol{H}^{p\times p}$ and $\lfloor \cdot \rfloor$ is the floor function. The depth-wise convolution extracts feature information channel by channel before mergers.

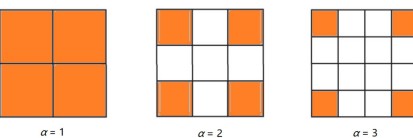

Figure 4: Convolution Kernels with Different Void Rates.

Table 1: Dataset Statistics for Link Prediction.

| Dataset | Entity | Relation | Degree | Train Set | Val. Set | Test Set |
|---------|--------|----------|--------|-----------|----------|----------|
| WN18RR | 40943 | 11 | 2.2/3.6 | 86,835 | 3034 | 3134 |
| FB15k-237 | 14541 | 237 | 19.7/30 | 272,115 | 17535 | 20466 |
| YAGO3-10 | 123182 | 37 | 9.6/8.7 | 1,079,040 | 5000 | 5000 |

Wang et al. [2021] suggest that while single-size kernels benefit from parameter sharing and low computing overhead, their receptive fields are limited. On the contrary, the multi-size circular convolution with the attention mechanism could better extract critical features, so in this study the atrous convolution is employed. with the equivalent kernel size defined as

$$p' = p + (p-1)(\alpha - 1), \qquad (9)$$

where $p$ is the size of a standard kernel, and $\alpha$ is the void rate. Holes are filled with 0, so the receptive field is enlarged with same number of parameters and same computational cost. Convolution kernels with different void rates are shown in Figure 4. . Given the number of kernels of each size C and feature matrix is $\pi(\mathcal{P}_k)$, features extracted by the $j$th ($j = 1, 2, , C$) kernel of the $i$th ($i = 1, 2, 3$) size are denoted as

$$\boldsymbol{V} = \mathrm{f}\left(\pi(\mathcal{P}_k)\mathrm{conv}(\boldsymbol{\omega}_i^j) + \boldsymbol{b_i}\right), \qquad (10)$$

where $\boldsymbol{b_i}$ is the bias and $\boldsymbol{V}_1, \boldsymbol{V}_2, \boldsymbol{V}_3 \in \boldsymbol{R}^{C \times 2m \times n}$.

In order to reduce noise and highlight key features, the attention module is introduced to adaptively adjust the weights of features from various kernels. With convolution the score function is modified, denoted as

$$\phi(\boldsymbol{h}, \boldsymbol{r}, \boldsymbol{t}) = \| \mathrm{conv}(\boldsymbol{h}, \boldsymbol{r}) \circ (\boldsymbol{h} \otimes \boldsymbol{r}) - \\ \mathrm{conv}(\boldsymbol{r}, \boldsymbol{t}) \circ \boldsymbol{t} \| \qquad . \qquad (11)$$

where $\circ$ denotes the Hadamard product and $\otimes$ denotes the Hamilton product. The loss gradient of entity/relation embeddings could propagate bi-directionally through the convolution or hypercomplex multiplications.

# 4 EXPERIMENTS

## 4.1 LINK PREDICTION

Given an entity and a relation, the missing entity is predicted. The higher the ranking of correct triples in the candidate set are, the stronger the prediction capability of the model is. Mean Reciprocal Rank (MRR) and the proportion of correct entities / triples in the top $N$ candidates (Hits@$N$,$N = 1, 3, 10$) are selected as metrics. The higher the score, the better. Bernoulli method [Wang et al., 2014] is adopted to randomly replace entities to create invalid triples. Filtered strategy is employed [Bordes et al., 2013]. Head and tail predictions are regarded as one task and the scores are combined.

Experiments are conducted on three benchmark datasets: WN18RR [Dettmers et al., 2018], FB15k-237 [Toutanova and Chen, 2015] and YAGO3-10 [Mahdisoltani et al., 2014]. WN18RR and FB15k-237 remove inverse relations to fix the high-score flaw. The relations in the YAGO3-10 dataset are mostly descriptive attributes about human. Some relations are with hierarchical structure, such as *hypernym* (WN18RR), *part-of* (FB15k-237) and *playsFor* (YAGO3-10). Dataset statistics are shown in Table 1, in which the degrees reflect the relational complexity of the datasets [Dettmers et al., 2018].

The following models are used as baselines: 1. TransE: Results from Ruffinelli et al. [2019]. 2. RotatE: Results from Sun et al. [2019]. 3. Rotate3D: Results from Gao et al. [2020]. 4. DistMult, ComplEx and ConvE: Results from Dettmers et al. [2018]. 5. ComplEx-N3: Results from Lacroix et al. [2018]. 6. QuatE: Results from Zhang et al. [2019]. We also make our own implementation and run on YAGO3-10, etc. (Codes are released on https://gitee.com/tkgc/APAC.) 7. ROTE/ATTE: embedding models in hyperbolic spaces, ATTE combining rotation and reflection while ROTE only containing rotation. Results from Chami et al. [2020]. 8. TuckER: a SOTA semantic model with TuckER Decomposition. Results from Balažević et al. [2019b]. 9. CoKE: A SOTA path model employing Transformer to encode semantics. Results from Wang et al. [2019].

The Training details are in Appendix B. Results are shown in Table 2 taking a 5-time average. Results in bold indicate the best performance while those in italics are the second. $\mathrm{APAC_q}$ and $\mathrm{APAC_o}$ denote quaternion and octonion embedding respectively. It is obvious that the overall performance of APAC is better than mainstream models and $\mathrm{APAC_q}$ is significantly better than $\mathrm{APAC_o}$ at many indicators.

On WN18RR, the simplest dataset, Rotate3D and QuatE achieve best results while $\mathrm{APAC_q}$ secures the highest MRR and good Hits@1 Score. On the complex dataset FB15k-237, the advantages of $\mathrm{APAC_q}$ are more clear with highest MRR and Hits@1, 3. the Hits@1 score is 3.3% higher than that of QuatE while the Hits@10 score is catching up. On Yago3-10, the largest dataset, there is no comparison with Rotate3D (no available code), but $\mathrm{APAC_q}$

Table 2: Link Prediction Results.

| Model | WN18RR | | | | FB15k-237 | | | | YAGO3-10 | | | |
|---|---|---|---|---|---|---|---|---|---|---|---|---|
| | MRR | Hits@1 | 3 | 10 | MRR | Hits@1 | 3 | 10 | MRR | Hits@1 | 3 | 10 |
| TransE | 0.228 | - | - | 0.520 | 0.313 | - | - | 0.497 | - | - | - | - |
| RotatE | 0.476 | 0.428 | 0.492 | 0.571 | 0.338 | 0.241 | 0.375 | 0.533 | 0.495 | 0.402 | 0.550 | 0.670 |
| Rotate3D | *0.489* | 0.442 | **0.505** | **0.579** | 0.347 | 0.250 | 0.385 | 0.543 | - | - | - | - |
| DistMult | 0.430 | 0.390 | 0.440 | 0.490 | 0.241 | 0.155 | 0.263 | 0.419 | 0.340 | 0.240 | 0.380 | 0.540 |
| ComplEx | 0.440 | 0.410 | 0.460 | 0.510 | 0.247 | 0.158 | 0.275 | 0.428 | 0.360 | 0.260 | 0.400 | 0.550 |
| ComplEx-N3 | 0.470 | - | - | 0.540 | 0.350 | - | - | 0.540 | 0.490 | - | - | *0.680* |
| QuatE | 0.482 | 0.436 | *0.499* | *0.572* | *0.366* | 0.271 | *0.401* | **0.556** | 0.502 | 0.428 | 0.543 | 0.674 |
| ROTE | 0.463 | 0.426 | 0.477 | 0.529 | 0.307 | 0.220 | 0.337 | 0.482 | 0.381 | 0.295 | 0.417 | 0.548 |
| ATTE | 0.456 | 0.419 | 0.471 | 0.526 | 0.311 | 0.223 | 0.339 | 0.488 | 0.374 | 0.290 | 0.410 | 0.537 |
| TuckER | 0.470 | 0.443 | 0.482 | 0.526 | 0.358 | 0.266 | 0.394 | 0.544 | - | - | - | - |
| CoKE | 0.484 | **0.450** | 0.496 | 0.553 | 0.364 | *0.272* | 0.400 | *0.549* | - | - | - | - |
| ConvE | 0.460 | 0.390 | 0.430 | 0.480 | 0.316 | 0.239 | 0.350 | 0.491 | *0.520* | *0.450* | **0.560** | 0.660 |
| APAC$_q$ | **0.501** | *0.447* | 0.487 | 0.535 | **0.378** | **0.280** | **0.407** | 0.548 | 0.518 | **0.461** | *0.558* | **0.696** |
| APAC$_o$ | 0.479 | 0.435 | 0.488 | 0.539 | 0.353 | 0.269 | 0.384 | 0.511 | **0.527** | 0.422 | 0.546 | 0.620 |

achieves highest or close to highest scores at all metrics, Hits@1 score 7.7% higher than that of QuatE and Hits@10 score higher than that of QuatE. APAC$_o$ performs well at MRR and Hits@3. Comparison with Rotate3D and QuatE demonstrates that APAC holds strong learning ability for complex relational patterns which may come from the integration of path semantics or the depth-wise atrous circular convolution combined with the attention mechanism. Compared with TransE, DistMult, ComplEx, ROTE, ATTE and TuckER, APAC$_q$ and APAC$_o$ perform better indicating the effectiveness of hypercomplex embedding. APAC$_q$ also holds certain advantages over CoKE, the SOTA context semantics model, verifying the forces of hypercomplex embedding and feature extraction methods. ConvE does not perform great due to the simple reshaping strategy. Follow-up experiments focus on APAC$_q$.

## 4.2 PATH QUERY

To verify model capabilities for modeling the composition pattern path query (multi-hop reasoning) is carried out following Rotate3D. Given the starting entity $h$ and the path $p$, entities that $h$ can reach via $p$ are predicted and ranked. Two datasets provided by Guu et al. [2015] are employed, coming from WordNet and Freebase respectively. Dataset statistics are shown in Table 3. The same settings including the negative sampling and filtering strategy by Gao et al. [2020] are adopted. The average quantile (MQ) and Hits@10 are used as metrics. The higher the score, the better. Two training strategies are employed: only triples (denoted as Single) and all paths (Comp).

Best performance is achieved when the embedding dimension $d = 500$, batch size 512, learning rate $lr = 0.0005$, margin $\gamma = 8$ and other parameters same as on YAGO3-10.

Compare APAC with Bilinear [Guu et al., 2015], TransE, CoKE, RotatE and Rotate3D under the Single strategy, with ROP [Yin et al., 2018], using RNN to model paths), CoKE, RotatE and Rotate3D under the Comp strategy. Relevant results are from Gao et al. [2020] and the results are shown in Table 4, Table 5. It can be seen that except for the MQ score on WordNet, APAC$_q$-Single wins on all metrics in its group. APAC$_q$-Comp also achieves the highest or second highest scores for each indicator, and scores higher than APAC$_q$-Single, which is another proof APAC$_q$ possesses the learning ability for the composition pattern and the ability to integrate path semantics. CoKE relies heavily on contexts, so it's a draw for CoKE and APAC$_q$-Comp under the Comp strategy. However, APAC$_q$-Comp performs much better under the Single strategy.

## 4.3 APPLICATION ON INDUSTRY DATASET

Domain-specific KGs are helpful for promoting knowledge application and the industry development. Apply our model to a biomedical dataset ogbl-biokg [1] containing 5 entity types including diseases, drugs, side effects, proteins and their functions as well as 51 relation types. The statistics is shown in Table 6. ogbl-biokg is collected from diversified sources with complex relation modes and broad confidence differences for facts, challenging models for extracting relation features and modeling knowledge uncertainty (another future plan). Random divisions of the train/val./test sets are made with proportions 94%, 3% and 3% respectively. Since the entity relations are rather dense and simply replacing the head or tail entity probably brings false negatives, replace the head and the tail entities at the same time to generate invalid samples with the ratio set to 1:1.

Best performance is achieved when the embedding dimen-

---

[1]https://ogb.stanford.edu/docs/linkprop/#ogbl-biokg

Table 3: Dataset Statistics for Path Query.

| Dataset | Entity | Relation | Train Set | Val. Set | Test Set | Train Paths | Val. Paths | Test Paths |
|---------|--------|----------|-----------|----------|----------|-------------|------------|------------|
| WordNet | 38551 | 11 | 110,361 | 2602 | 10462 | 2129539 | 11277 | 56477 |
| Freebase | 75043 | 13 | 316,232 | 5908 | 23733 | 6266058 | 27163 | 109557 |

Table 4: Path Query Results (Triple Training).

| Model | WordNet | | Freebase | |
|-------|---------|---------|----------|---------|
| | MQ | Hits@10 | MQ | Hits@10 |
| Bilinear-Single | 0.847 | 0.436 | 0.580 | 0.259 |
| TransE-Single | 0.837 | 0.138 | 0.862 | 0.454 |
| CoKE-Single | 0.731 | 0.157 | 0.730 | 0.367 |
| RotatE-Single | *0.937* | 0.479 | 0.833 | 0.453 |
| Rotate3D-Single | **0.941** | *0.494* | *0.894* | *0.547* |
| $APAC_q$-Single | 0.932 | **0.502** | **0.904** | **0.583** |

Table 5: Path Query Results (Path Training).

| Model | WordNet | | Freebase | |
|-------|---------|---------|----------|---------|
| | MQ | Hits@10 | MQ | Hits@10 |
| ROP-Comp | - | - | 0.907 | 0.567 |
| CoKE-Comp | 0.942 | *0.674* | **0.948** | **0.764** |
| RotatE-Comp | 0.947 | 0.653 | 0.901 | 0.601 |
| Rotate3D-Comp | *0.949* | 0.671 | 0.905 | 0.621 |
| $APAC_q$-Comp | **0.960** | **0.719** | *0.933* | *0.723* |

sion $d = 500, 1000$, batch size 512 and the learning rate $lr = 0.0001$, other parameters same as on YAGO3-10.

Compare $APAC_q$ with TransE, DistMult, RotatE, QuatE, PairRE [Chao et al., 2020] and AutoSF+ [Zhang et al., 2021b]. The latter two are specially designed for modeling complex relations. PariRE introduces relationship-specific pair vectors for representation while in AutoSF+ an adaptive score function is proposed, pruning the search spaces with filters and predictors and replacing the greedy algorithm in AutoSF [Zhang et al., 2020]with Evolutionary Search.

Results are shown in Table 7. $APAC_q$ performs best with slight differences between dimensions 500 and 1000, showing strong feature extraction capability under low dimensions. Translation / rotation models that only extract shallow features do not perform great and increasing dimensions does not bring obvious improvement. A similar situation occurs with QuatE. Compared with the semantic models, $APAC_q$ holds obvious advantages. Compared with PariRE and AutoSF+, $APAC_q$ upgrades performance without adding embedding or expanding the search spaces, which verifies the effectiveness of path semantic integration and our convolution framework.

It is worth noting that although additional path calculation and convolution operations are introduced, with the phased parallel training strategy, it only takes 1.5 hours for $APAC_q$ to surpass QuatE's best performance under dimension 500, which is only about 1/5 of the training time for the latter, indicating that our model improves feature extraction capability while boosting computing efficiency.

## 5 CONCLUSION

Compared with embedding models in real / complex spaces, a knowledge representation model with stronger expressivity and feature extraction capabilities is proposed with hypercomplex embedding, path semantic integration combining fast quaternion rotation blending and the attention mechanism, and the depth-wise atrous circular convolution. Low computational cost of quaternion multiplication, parameter sharing in CNN and phased parallel training strategy ensure rapid taking effect of the model on large datasets. Future work includes further improving on learning complex composition modes, applying Kronecker product to expand embedding spaces with high efficiency [Zhang et al., 2021a] and modeling knowledge uncertainty / time validity, etc.

## A PROOF AND ILLUSTRATION OF THEOREM 1

Proof. For $q_1 \otimes q_2 \, q_2 \otimes q_1$, bisect the angles respectively, $q_1' = q_1^{\frac{1}{2}}, q_2' = q_2^{\frac{1}{2}}$, resulting in $q_1^{\frac{1}{2}} * q_2^{\frac{1}{2}} * q_1^{\frac{1}{2}} * q_2^{\frac{1}{2}}$ and $q_2^{\frac{1}{2}} * q_1^{\frac{1}{2}} * q_2^{\frac{1}{2}} * q_1^{\frac{1}{2}}$. Parts of the two calculations are the same. Further $k$-sect $q_1, q_2$. The greater the value of $k$ is, the higher the proportion of same calculations is. When $k \to \infty$, the middle parts of the two calculations converge while the head and tail parts tend to be close to unit quaternions with weakening influence, leading to stable results. According to Alexa [2002], the limit exists, so we have

$$\lim_{k \to \infty} \left( q_1^{\frac{1}{k}} * q_2^{\frac{1}{k}} \right)^k = \lim_{k \to \infty} \left( q_2^{\frac{1}{k}} * q_1^{\frac{1}{k}} \right)^k. \quad (12)$$

For the Trotter product formula

$$e^{A+B} = \lim_{N \to \infty} \left( e^{\frac{A}{N}} * e^{\frac{B}{N}} \right)^N, \quad (13)$$

Table 6: Dataset Statistics for ogbl-biokg.

| Dataset | Diseases | Drugs | Side Effects | Proteins | Functions | Total Entities | Train Set | Val. Set | Test Set |
|---------|----------|-------|--------------|----------|-----------|----------------|-----------|----------|----------|
| ogbl-biokg | 10687 | 10533 | 9969 | 17499 | 45085 | 93773 | 4.76M | 162k | 162k |
| | 11.40% | 11.23% | 10.63% | 18.66% | 48.08% | 100% | 94% | 3% | 3% |

Table 7: Dimensions and MRR on ogbl-biokg.

| Model | Dimension | MRR |
|-------|-----------|-----|
| TransE | 2000 | 0.7452 |
| RotatE | 1000 | 0.7989 |
| DistMult | 2000 | 0.8043 |
| ComplEx | 1000 | 0.8095 |
| QuatE | 500 | 0.7712 |
| QuatE | 1000 | 0.7954 |
| PairRE | 2000 | 0.8164 |
| AutoSF+ | 1000 | 0.8309 |
| AutoSF+ | 2000 | 0.8320 |
| $APAC_q$ | 500 | *0.8526* |
| $APAC_q$ | 1000 | **0.8578** |

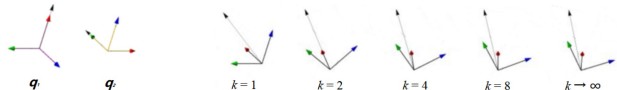

Figure 5: Rotations under Different $k$-sections.

replace $e^{\frac{A}{N}}, e^{\frac{B}{N}}$ with $\boldsymbol{q_1^{\frac{1}{k}}}, \boldsymbol{q_2^{\frac{1}{k}}}$. With $e^{\log \boldsymbol{q}} = \boldsymbol{q}$, we have

$$\lim_{n \to \infty} \left( \boldsymbol{q_1^{\frac{1}{k}}} \otimes \boldsymbol{q_2^{\frac{1}{k}}} \right)^k = e^{\log \boldsymbol{q_1} + \log \boldsymbol{q_2}} \tag{14}$$

(The formal proof of isomorphism between quaternions and matrices is saved for future work). It can be seen that the limit operation is equivalent to find the sum of the logarithms of the two quaternions and compute the exponentiation of the result. The calculation cost is constant. Further extend such operation to a quaternion sequence and we get (4). When the special case of coaxial rotational blending (on the same plane) occurs, the result could be seen as a quaternion formed by adding all rotational angles together.

Illustration Rotations represented by quaternions $q_1, q_2$ under different $k$-sections are shown in Figure 5. . The axes in 3D spaces are in red, blue and green respectively and the black arrow is the vector calculated by the axis and the angle. It can be seen that when $k = 8$, the result is close to the result when $k \to \infty$.

## B MODEL TRAINING DETAILS

Experiments are conducted on a Lenovo SR590 server with the hardware configuration including 20 core Xeon * 2 (CPU), 16G * 8 Memory, 1.2TB * 3 SAS disks (in RAID5 mode) and Tesla P100 * 2 (computing cards).

Adam optimizer (Adaptive Estimates of Lower-Order Moments) is adopted and optimal parameters are determined with Grid Search. The hyperparameter pool and optimal parameters could be found on the project homepage. Dropouts are added before and after convolution and after the full connection layer, numbered 1, 2 and 3 respectively. Following Gao et al. (2020), relation-specific biases are applied. The batch normalization strategy is employed to reduce the scaling effect caused by hypercomplex multiplications and control the normalizing rate. It is found that with the batch normalization models converge faster and perform more stable than with the unit quaternion. Early stop strategy is activated when the MRR increase in the last 10 epochs on Val. Set is less than 10-2. In other experiments the optimizer and training strategies are the same unless declared differently.

**Author Contributions**

Ms. Gao and Ms. Shi help with theretical deduction and verification of the combination of the atrous and circular convolution. Ms. Zhou and Mr. Husen oversee the overall framework and project implementation.

**Acknowledgements**

The study is supported by the provincial research project in Fujian (FJJKCG20-402).

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
