# OpenReview forum: "Knowledge Representation Combining Quaternion Path Integration and Depth-wise Atrous Circular Convolution"
_auai.org/UAI/2022/Conference — UAI 2022 Poster_

### Official Review · Reviewer_GZSU · 2022-04-10

**Q2(1) Originality/Novelty:** 2
**Q2(2) Significance/Impact:** 2
**Q2(3) Correctness/Technical Quality:** 2
**Q2(6) Clarity Of Writing:** 2
**Q6 Overall Score:** 3
**Q8 Confidence In Your Score:** 4

**Q1 Summary And Contributions:**

The paper proposed APAC, a knowledge representation model based on the non-commutative group.
For the link prediction task, the model minimizes the distance between the tail entity and the product of the head entity and the relation in the quaternion/octonion space. The product grows to multiplication over all path entities and relations for the path query task.
To better extract features, the hyperspace representations are processed by a convolution layer before feeding into the loss function.

**Q2 Assessment Of The Paper:**

More detailed information regarding each of these aspects is given below:

**Q2(4) Quality Of Experiments (Optional):**

2: Fair: The experimental evaluation is weak: important baselines are missing, or the results do not adequately support the main claims.

**Q2(5) Reproducibility:**

2: Fair: Key resources (e.g., proofs, code, data) are unavailable but key details (e.g., proof sketches, experimental setup) are sufficiently well-described for an expert to confidently reproduce the main results.

**Q3 Main Strengths:**

1. The paper proposed an efficient way to multiply a sequence of quaternions together.
2. The proposed method demonstrated SOTA performance on most metrics in link prediction and path query tasks.


**Q4 Main Weakness:**

1. The paper lacks scientific contributions. It seems like the paper simply combines the Non-Alebian knowledge model (Chami et al., 2020, Gao et al., 2021) and atrous convolution (Wang et al., 2021).
2. The experimental result lacks interpretation. For example, it is unclear why the octonion model outperforms the quaternion model on most metrics.
3. The paper lacks substance and is slightly over 6 pages.


**Q5 Detailed Comments To The Authors:**

Comments, Suggestions, And Typos:
1. "since" is spelled as "sine" in the second last paragraph of section 2.
2. Missing citation in the first paragraph of section 4.



**Q7 Justification For Your Score:**

not much contribution or substance

**Q9 Complying With Reviewing Instructions:**

1: Yes.

---

### Official Review · Reviewer_5yfy · 2022-04-11

**Q2(1) Originality/Novelty:** 2
**Q2(2) Significance/Impact:** 3
**Q2(3) Correctness/Technical Quality:** 3
**Q2(6) Clarity Of Writing:** 2
**Q6 Overall Score:** 5
**Q8 Confidence In Your Score:** 3

**Q1 Summary And Contributions:**

This paper combines quaternion path integration and depth-wise atrous circular convolution to enhance the expressivity of embedding. It uses quaternion algebra and a Hamilton group with the smallest order to learn multiple relation embeddings. A fast multiplicative calculation method is proposed to rapidly merge feature of relational paths with the attention mechanism. Additionally, a depth-wise atrous circular convolution framework is applied to capture rich features of complex relations.

**Q10 Ethical Concerns (Optional):**

No.

**Q2 Assessment Of The Paper:**

More detailed information regarding each of these aspects is given below:

**Q2(4) Quality Of Experiments (Optional):**

3: Good: The experimental evaluation is adequate, and the results convincingly support the main claims.

**Q2(5) Reproducibility:**

3: Good: Key resources (e.g., proofs, code, data) are available and key details (e.g., proofs, experimental setup) are sufficiently well-described for competent researchers to confidently reproduce the main results.

**Q3 Main Strengths:**

The APAC model integrates hypercomplex embedding, path semantic integration combining fast quaternion rotation blending and the attention mechanism that has stronger expressivity and feature extraction capabilities. The results of link prediction and path query show that the APAC achieves SOTA performance on most datasets.

**Q4 Main Weakness:**

1. The motivation of this paper is unclear. It would be better to explain why the three techniques (hypercomplex embedding, path semantic integration, Depth-wise Atrous Circular Convolution) used in the paper are necessary for enhancing the expressivity of KG embeddings.
2. The presentation of the paper would need improvement. The font size in Table 2, Table 3 and Table 6 is too small.
3. Probably due to the page limitation, many details are missing, e.g., case studies and the embedding distribution of different methods in the representation space.


**Q5 Detailed Comments To The Authors:**

The main idea of this paper is combining three methods to capture the rich features of knowledge graph. The experiment results show the proposed model outperforms the mainstream methods. However, the major concern with this paper is the motivation. In the introduction section, authors introduce existing embedding models, but the problems with these models are not analyzed. The paper only employs three techniques to KG representations, with no explantations on why these techniques can improve the representation ability of embeddings.

**Q7 Justification For Your Score:**

See above.

**Q9 Complying With Reviewing Instructions:**

1: Yes.

---

### Official Review · Reviewer_fykf · 2022-04-15

**Q2(1) Originality/Novelty:** 3
**Q2(2) Significance/Impact:** 3
**Q2(3) Correctness/Technical Quality:** 3
**Q2(6) Clarity Of Writing:** 3
**Q6 Overall Score:** 6
**Q8 Confidence In Your Score:** 4

**Q1 Summary And Contributions:**

This paper aims at improving knowledge representation with hypercomplex embedding. It is enabled by using path semantic integration with fast quaternion rotation blending and attention mechanism plus depth-wise atrous circular convolution. This paper also provides a proof for rapid feature mergers of relational paths. Experiments on link prediction and path query demonstrate the effectiveness of the proposed method.

**Q2 Assessment Of The Paper:**

More detailed information regarding each of these aspects is given below:

**Q2(4) Quality Of Experiments (Optional):**

3: Good: The experimental evaluation is adequate, and the results convincingly support the main claims.

**Q2(5) Reproducibility:**

3: Good: Key resources (e.g., proofs, code, data) are available and key details (e.g., proofs, experimental setup) are sufficiently well-described for competent researchers to confidently reproduce the main results.

**Q3 Main Strengths:**

S1: A theoretical method with proof is proposed in hypercomplex space to improve knowledge representation.
S2: Several techniques are presented to make the proposed method work.
S3: The experiments seem quite good.

**Q4 Main Weakness:**

W1: The writing of this paper needs much improvement. For example, equations (3) and (11) are not easy to understand. Figure 1 and Figure 3 need to be modified to make it clear. Tables are not the same size and several tables are very small making it less readable.
W2: This paper introduces several techniques and the relationship should be clearly described.

**Q5 Detailed Comments To The Authors:**

This paper aims at improving knowledge representation with hypercomplex embedding. It is enabled by using path semantic integration with fast quaternion rotation blending and attention mechanism plus depth-wise atrous circular convolution. This paper also provides a proof for rapid feature mergers of relational paths. Experiments on link prediction and path query demonstrate the effectiveness of the proposed method.

Strengths:
S1: A theoretical method with proof is proposed in hypercomplex space to improve knowledge representation.
S2: Several techniques are presented to make the proposed method work.
S3: The experiments seem quite good.

Weaknesses:
W1: The writing of this paper needs much improvement. For example, equations (3) and (11) are not easy to understand. Figure 1 and Figure 3 need to be modified to make it clear. Tables are not the same size and several tables are very small making it less readable.
W2: This paper introduces several techniques and the relationship should be clearly described.

**Q7 Justification For Your Score:**

A theoretical method with proof is proposed in hypercomplex space to improve knowledge representation.

**Q9 Complying With Reviewing Instructions:**

1: Yes.

---

### Decision · Program_Chairs · 2022-05-15

**Decision:**

Accept (Poster)

**Comment:**

Meta Review: The negative review is about 1) unclear writing and 2) lack of experimental analysis. Both of them are addressed by the authors in response. AC agrees that the paper requires substantial revision before publication. AC recommends acceptance